# Chymase Inhibition Attenuates Kidney Fibrosis in a Chronic Mouse Model of Renal Ischemia–Reperfusion Injury

**DOI:** 10.3390/ijms26083913

**Published:** 2025-04-21

**Authors:** Sakura Kure, Hiroe Toba, Denan Jin, Akira Mima, Shinji Takai

**Affiliations:** 1Department of Innovative Medicine, Graduate School of Medicine, Osaka Medical and Pharmaceutical University, Takatsuki-City 569-8686, Osaka, Japan; ompu71123002@s.ompu.ac.jp (S.K.); shinji.takai@ompu.ac.jp (S.T.); 2Department of Nephrology, Osaka Medical and Pharmaceutical University, Takatsuki-City 569-8686, Osaka, Japan; akira.mima@ompu.ac.jp; 3Department of Clinical Pharmacology, Division of Pathological Sciences, Kyoto Pharmaceutical University, 1 Misasagi Shichono-cho, Yamashina-ku 607-8412, Kyoto, Japan; toba@mb.kyoto-phu.ac.jp; 4Department of Pharmacology, Osaka Medical and Pharmaceutical University, Takatsuki-City 569-8686, Osaka, Japan

**Keywords:** chymase, mast cells, chymase inhibitor, angiotensin II, transforming growth factor (TGF)-β1

## Abstract

Although various factors contribute to the transition from acute kidney injury (AKI) to chronic kidney disease (CKD), no clinically effective pharmacological treatment has been established. We investigated whether chymase inhibition is effective in preventing renal fibrosis, a key process in the transition from AKI to CKD. Male BALB/c mice were subjected to unilateral ischemia-reperfusion (I/R) injury, and TY-51469, a chymase-specific inhibitor, was administered intraperitoneally at a dose of 10 mg/kg/day for 6 weeks. The 45 min ischemic period followed by 6 weeks of reperfusion resulted in severe renal atrophy. Renal fibrosis was particularly pronounced in the transition region between the cortex and medulla in placebo-treated mice. The expression of mouse mast cell protease 4 (MMCP-4, a mouse chymase) mRNA, the number of chymase-positive mast cells, and fibrosis-related factors, such as transforming growth factor (TGF)-β1 and collagen I, were all significantly increased in I/R-injured kidneys. However, treatment with TY-51469 significantly suppressed fibrosis formation, along with the inhibition of renal chymase and TGF-β1 expression. These findings suggest that chymase inhibition may be a potential therapeutic strategy for preventing the transition from AKI to CKD by reducing fibrosis.

## 1. Introduction

Acute kidney injury (AKI) is a significant risk factor for chronic kidney disease (CKD) and end-stage renal disease (ESRD). Various common pathological conditions can lead to AKI, including hemorrhage, fluid loss, circulatory failure, cardiovascular surgery, shock, and transplant surgery, resulting in a rapid decline in kidney function [1]. For a long time, AKI was considered a temporary condition, and it was believed that patients who recovered from AKI through medical treatment would not experience long-term complications. However, recent evidence suggests that AKI plays a crucial role in the progression to CKD. For example, even though AKI patients successfully recover from the acute phase and are discharged from the hospital, studies have reported that up to 50% of them develop CKD within a few years [2]. This suggests that kidney damage caused by AKI might persist despite aggressive treatment. Indeed, even when serum creatinine levels return to baseline, structural damage to the kidney can persist for a long time, potentially leading to the development of CKD [3,4,5]. Although the mechanisms underlying the transition from AKI to CKD are not yet fully understood, factors such as endothelial dysfunction [6,7], interstitial inflammation [8,9], fibrosis [10,11], and tubular epithelial injury [12,13] are thought to contribute to this progression. Endothelial dysfunction can lead to oxygen deficiency in kidney tissue, resulting in mitochondrial dysfunction in various cell types and damage to renal tubular epithelial cells, ultimately triggering sterile interstitial inflammation [14]. Simultaneously, repair mechanisms are activated in response to this inflammation. However, excessive repair following AKI can lead to renal fibrosis, further damaging the normal kidney structure and promoting the progression from AKI to CKD [9].

Recent studies have highlighted possible molecular mechanisms involved in the inflammatory and fibrotic processes following AKI. For example, inflammatory mediators such as tumor necrosis factor-alpha (TNF-α), interleukin-1 beta (IL-1β), and interleukin-6 (IL-6) are produced by the renal tubular epithelium after ischemia-reperfusion (I/R) injury [15,16,17]. Additionally, a significant increase in transforming growth factor-beta 1 (TGF-β1), a key mediator of fibrosis, has been observed in the cytoplasm of interstitial cells in mouse models of I/R injury [18]. These findings indicate that the activation of both inflammation-related factors and fibrosis-related mediators is critical in the transition from AKI to CKD. Although some causative factors in the transition from AKI to CKD after I/R injury in animal models have been identified, effective pharmacological therapies for preventing this progression have not yet been established in clinical practice.

Recently, mast cell-derived chymase, a novel component of the renin–angiotensin system (RAS), has attracted significant attention. Under certain conditions, such as ischemia or inflammation, chymase is released through degranulation and functions similarly to angiotensin-converting enzyme (ACE), cleaving angiotensin I (Ang I) into angiotensin II (Ang II) [19]. In addition to its role in Ang II generation, chymase can also activate latent TGF-β1, converting it into its active form [20]. Ang II is a well-known pro-inflammatory molecule involved not only in hypertension but also in fibrosis of various organs. Given these unique properties of chymase, we investigated its expression in the kidney following I/R injury and evaluated whether long-term treatment with TY-51469 could prevent renal fibrosis in a mouse model of unilateral renal I/R injury. TY-51469 is a chymase-specific inhibitor that has been shown in animal studies to be highly effective in treating albuminuria due to diabetic nephropathy [21], lipopolysaccharide/d-galactosamine-induced acute liver failure [22], and tetrachloride-induced liver fibrosis [23].

## 2. Results

### 2.1. Procedure for Creating the I/R Model and Subsequent Changes in the Kidney Weight (KW)-to-Body Weight (BW) Ratio

Figure 1A shows a photograph of the left kidney of an anesthetized mouse after 45 min of renal ischemia. Renal ischemia was induced by blocking both arterial and venous blood flow using an arterial clamp on the left renal pedicle. As seen in Figure 1A, the renal surface turned dark purple 45 min after pedicle clamping, indicating stagnant blood flow. However, a few seconds after removing the clamp, the kidney turned light red (Figure 1B), signifying the restoration of blood flow.

As shown in Figure 2D, the BW of each experimental group was approximately the same before the start of the experiments. However, BW increased to some extent in all the experimental groups 6 weeks after surgery. Figure 2A shows representative photographs of the right and left kidneys 6 weeks after the sham operation. As seen in the photograph, there was not much difference in size between the two kidneys. Similarly, the KW/BW ratio showed no significant difference between the left and right kidneys 6 weeks after the sham operation (Figure 2E). However, six weeks after the I/R injury, the ischemic left kidney was significantly shrunken compared to the contralateral and sham-operated left kidneys (Figure 2B). When KW was adjusted for BW, the KW/BW ratio of the ischemic left kidney was significantly lower than that of the contralateral and sham-operated kidneys (Figure 2E). Interestingly, the contralateral healthy right kidney in the I/R mice seemed to undergo compensatory hypertrophy, resulting in a significantly higher KW/BW ratio compared to the sham-operated right kidney (Figure 2E). As seen in Figure 2C,E, the KW/BW ratio of the left I/R kidney in TY-51469-treated mice was similar to that of the placebo-treated group. However, the compensatory mechanisms in TY-51469-treated mice appeared to be less severe, as indicated by the fact that the KW/BW ratio of the right kidney in these mice did not differ significantly from that of the sham-operated right kidney (Figure 2E).

### 2.2. Characteristics of Renal Fibrosis After I/R

As shown in Figure 3A, Azan–Mallory staining of the sham-operated kidney revealed normal histology with no significant renal fibrosis. As is well known, the Azan–Mallory staining method stains collagen fibers within fibrous connective tissue blue. Since collagen fibers are commonly distributed around blood vessels, these areas typically appear blue with Azan–Mallory staining, even in normal kidneys. In Figure 3A, no obvious blue staining was observed in the sham-operated kidney, aside from the tissue surrounding blood vessels.

However, in the 6-week I/R injured placebo-treated kidneys, there were remarkably more blue-stained areas at the transition region between the cortex and medulla (Figure 3B), and the calculated fibrotic area in the placebo-treated group was significantly larger compared to that in the sham-operated group (Figure 3D). On the other hand, TY-51469 treatment, which commenced one day after the I/R injury, significantly reduced fibrosis in the I/R kidney (Figure 3C,D).

### 2.3. Changes in Chymase-Positive Mast Cells in Renal Fibrosis Areas After I/R

Figure 4A shows representative toluidine blue staining for the identification of mast cells in a sham-operated kidney 6 weeks after surgery. Toluidine blue staining is a simple and traditional method used to identify mast cells. The cytoplasm of mast cells exhibits metachromatic staining, appearing red-purple, which makes them easily distinguishable from other cell types. As seen in this section, few mast cells were observed in the non-injured healthy kidney. However, mast cells accumulated in the fibrotic region of the kidney 6 weeks after I/R (Figure 4B), with a significant increase in the mean number of mast cells in the I/R-injured kidney—by dozens of times—compared to sham-operated non-injured kidneys (Figure 4D). Figure 4C shows mast cell distribution in the I/R kidney from a mouse treated with TY-51469 for 6 weeks. As indicated in Figure 4D, although the number of mast cells in the TY-51469-treated I/R kidney was still significantly higher than that in the sham-operated group, TY-51469 treatment significantly reduced mast cell accumulation compared to the non-treated placebo group (Figure 4D).

Figure 5A shows representative chymase (mouse mast cell protease 4, MMCP-4) immunostaining in a sham-operated kidney 6 weeks after surgery. MMCP-4 is a mouse chymase typically stored in mast cells. In the present study, we prepared serial sections of kidney tissue to examine the expression pattern of MMCP-4 and to identify the cell types expressing MMCP-4 following I/R injury. Similar to Figure 4A (toluidine blue staining), few chymase-positive cells were observed in the non-injured healthy kidney (Figure 5A). However, chymase-positive cells increased dramatically in the fibrotic region of the kidney 6 weeks after I/R (Figure 5B), and their distribution pattern closely resembled that of the mast cells in Figure 4B. Since Figure 4B and Figure 5B present adjacent serial sections, these findings suggest that chymase is primarily expressed in mast cells. Compared to the sham-operated group, the number of chymase-positive cells significantly increased in I/R-injured kidneys (Figure 5D), while continued TY-51469 treatment for 6 weeks largely reduced chymase-positive cell expression (Figure 5C,D).

### 2.4. Changes in Gene Expression Levels of MMCP-4, TGF-β1, and Collagen I After I/R

Figure 6A shows MMCP-4 mRNA expression levels 6 weeks after surgery. As seen in the bar graph in Figure 6A, MMCP-4 expression in I/R-injured kidneys was significantly higher compared to non-injured sham-operated left kidneys and contralateral kidneys. However, MMCP-4 expression in I/R-injured kidneys was significantly suppressed by TY-51469 treatment. Fibrosis-related factors, such as TGF-β1 and collagen I, were also significantly elevated in I/R-injured kidneys compared to the sham-operated left kidney and the contralateral uninjured kidney. These elevations were significantly suppressed by TY-51469 treatment (Figure 6B,C).

### 2.5. Correlation Among Renal MMCP-4, TGF-β1, and Collagen I Gene Expression Levels After I/R

As seen in Figure 7A, the correlation coefficient (r) between MMCP-4 and TGF-β1 was 0.443, with a *p*-value of less than 0.05, indicating a close relationship between these factors. A significant correlation was also observed between the mRNA expression levels of MMCP-4 and collagen I, as well as between TGF-β1 and collagen I.

## 3. Discussion

Both unilateral and bilateral I/R injury models are commonly used to mimic clinical ischemic kidney injury in medical research. A certain duration of ischemia causes oxygen deprivation in kidney tissues, leading to damage in various kidney cells. While subsequent reperfusion restores the oxygen supply, the process might exacerbate kidney damage due to the excessive production of reactive oxygen species [24,25]. I/R injury is characterized by endothelial dysfunction, cell death via apoptosis and necrosis, and immune cell accumulation, ultimately leading to progressive renal fibrosis. This excessive repair mechanism disrupts the normal renal structure and gradually progresses to CKD [26,27]. In this study, we employed a unilateral renal I/R model in mice to investigate whether mast cell-derived chymase plays a role in kidney fibrosis over a 6-week observation period since the dual-kidney I/R model has a high mortality rate and is unsuitable for long-term analysis of fibrosis [28]. As shown in Figure 2E, no significant difference in the KW/BW ratio between the left and right kidneys was observed 6 weeks after the sham operation. However, in placebo-treated I/R mice, significant atrophy of the ischemic left kidney was noted. Interestingly, the KW/BW ratio of the contralateral, non-ischemic right kidney was significantly higher in the placebo-treated group compared to the sham group (Figure 2E), suggesting the occurrence of compensatory hypertrophy due to long-term dysfunction of the ischemic left kidney in the placebo group. Similarly, the ischemic left kidney in TY-51469-treated mice also showed atrophy 6 weeks after I/R. However, unlike in placebo-treated mice, compensatory hypertrophy of the contralateral right kidney was significantly reduced, suggesting that TY-51469, a chymase-specific inhibitor, might mitigate the pathological changes in the ischemic kidney and improve renal function post-I/R. As a matter of fact, we found that the fibrotic area in the ischemic left kidney of the placebo-treated group was significantly increased compared to the sham-operated kidney, with fibrosis predominantly localized at the transition region between the cortex and medulla (Figure 3B,D). In contrast, long-term TY-51469 treatment effectively suppressed fibrosis following I/R injury (Figure 3C,D). These findings suggest that mast cell-derived chymase activation plays a crucial role in fibrosis during the long-term period after I/R injury in mice. Our observation that the number of mast cells and MMCP-4-positive cells in the placebo-treated ischemic left kidney was significantly increased compared to the sham-operated, non-ischemic kidney (Figure 4 and Figure 5) supports this hypothesis. The distribution of both mast cells and MMCP-4-positive cells was also concentrated in the transition region between the cortex and medulla, where fibrosis was most prominent. Additionally, MMCP-4 mRNA expression was significantly elevated in the ischemic left kidney of the placebo-treated group compared to the sham-operated kidney. Since TY-51469 treatment suppressed both chymase-positive mast cell accumulation and MMCP-4 gene expression, which correlated with reduced fibrosis, it is likely that chymase activation indeed played a role in the pathogenesis of renal fibrosis in this model.

What was the mechanism for the increase in mast cells in the kidney after I/R injury in this study? Reportedly, mast cell-derived chymase enzymatically cleaves membrane-bound stem cell factor (SCF) to release the bioactive form of SCF [29]. Soluble SCF is known to be produced by many types of cells, such as fibroblasts and endothelial cells, and the enzymatically activated SCF produced by mast cell-derived chymase not only acts as a major chemotactic factor for mast cells and their progenitors but also elicits cell–cell and cell–substratum adhesion, facilitates their proliferation, and sustains the survival, differentiation, and maturation of mast cells [30]. In this study, although we did not investigate changes in SCF in renal tissue after I/R injury, the activation of chymase-dependent SCF signaling might have contributed to the observed increase in renal mast cells following I/R injury.

How does chymase activation contribute to fibrosis formation after I/R injury? Mice reportedly express several types of chymases, including MMCP-1, MMCP-2, MMCP-3, and MMCP-4, each with distinct enzymatic properties [31]. Among them, MMCP-4 is functionally considered the most similar to human chymase, particularly in its ability to generate Ang II [32]. Ang II is known to promote superoxide production via NADPH oxidase (NOX) activation [33], leading to the release of pro-inflammatory cytokines such as TNF-α, IL-1β, and IL-6 through nuclear factor-kappa B (NF-κB) activation [34,35]. In addition to Ang II generation, chymase can activate latent TGF-β1 into its active form [20]. Since TNF-α, IL-1β, and IL-6 are all elevated following renal I/R injury [24,25], and TGF-β1 is considered a key pro-fibrotic factor involved in the transition from acute kidney injury (AKI) to CKD [26,27], inhibition of MMCP-4 by TY-51469 likely reduces the production of both inflammatory and fibrotic mediators, such as TNF-α and TGF-β1. These inhibitory effects of chymase might have contributed to the attenuation of fibrosis in this mouse unilateral I/R injury model.

The acute management of AKI primarily involves symptomatic treatment of complications, such as pulmonary edema and hyperkalemia. Chymase inhibitors are unlikely to offer immediate benefits in this setting. Currently, no therapeutic agents have been definitively established to prevent the progression from AKI to CKD. While ACE inhibitors are clinically used in the treatment of diabetic nephropathy, they might also help suppress the transition from AKI to CKD by inhibiting Ang II production. In comparison, chymase inhibitors not only suppress Ang II generation but also inhibit the activation of profibrotic factors, such as TGF-β1. This dual mechanism suggests that chymase inhibitors might offer superior therapeutic efficacy compared to ACE inhibitors in attenuating the progression from AKI to CKD.

### Limitations

In the present study, increased expression of MMCP-4, TGF-β1, and collagen I was observed at the mRNA level. However, as mRNA expression does not necessarily correlate with protein expression, the relationship between the mRNA and protein levels of these molecules remains unclear. Further studies are needed to clarify whether the observed changes in gene expression are reflected at the protein level. Another limitation of this study is the exclusive focus on the 6-week endpoint. We recognize that an earlier assessment of chymase activity and inflammatory cytokine levels following I/R injury would have provided a more comprehensive understanding of chymase’s pathophysiological role. We plan to address these aspects in future studies.

## 4. Materials and Methods

### 4.1. Animals

Male BALB/c mice (SLC, Shizuoka, Japan), aged 6 weeks and weighing 18–22 g, were used in this study. All experimental procedures were conducted in accordance with the guidelines of Osaka Medical and Pharmaceutical University for medical experiments and were approved by the local institutional ethics committee (AM23-111). The mice were fed a standard diet, had free access to tap water, and were housed in a temperature-, humidity-, and light-controlled environment.

### 4.2. Creation of the Unilateral Renal I/R Injury Model

The renal I/R injury model was established under mixed anesthesia (0.3 mg/kg medetomidine, 4.0 mg/kg midazolam, and 5.0 mg/kg butorphanol tartrate). Briefly, mice were anesthetized via intraperitoneal injection (i.p.) of the mixed anesthetic and underwent midline incisions. The left renal pedicle was bluntly dissected, and an arterial clamp was applied for 45 min. During this period, the animals were maintained at a constant temperature (~37 °C) and kept well-hydrated. After 45 min of ischemia, the clamp was removed, the wounds were sutured, and the mice were allowed to recover. Sham-operated animals underwent the same surgical procedure without the ischemic insult.

### 4.3. Grouping and Sampling

The renal I/R injury model was induced in 12 mice, which were then divided into two groups: placebo-treated (*n* = 6) and TY-51469-treated (*n* = 6). TY-51469, a chymase-specific inhibitor, was administered intraperitoneally at a dose of 10 mg/kg once daily, starting one day after surgery and continuing until the end of the experiment. Sham operations were performed on an additional six mice. Six weeks after surgery, all 18 mice were sacrificed under pentobarbital overdose anesthesia, and both ischemic and normal kidneys were harvested and weighed. Each kidney was bisected longitudinally; one half was fixed in Carnoy’s solution for histological analysis, while the other half was frozen for biochemical examinations.

### 4.4. Real-Time Polymerase Chain Reaction (RT-PCR)

RT-PCR was performed to assess the expression of MMCP-4 (a mouse chymase), TGF-β1, and collagen I in renal tissues using previously described methods [21]. Briefly, total renal RNA was extracted using Trizol reagent (Life Technologies, Rockville, MD, USA) and dissolved in RNase-free water (Takara Bio Inc., Otsu, Japan). One microgram of total RNA was transcribed into complementary DNA (cDNA) using Superscript VIRO (Invitrogen, Carlsbad, CA, USA). mRNA levels were then quantified by RT-PCR using a Stratagene Mx3000P system (Agilent Technologies, San Francisco, CA, USA) and TaqMan fluorogenic probes. RT-PCR primers and probes for MMCP-4, collagen I, TGF-β1, and 18S ribosomal RNA (rRNA) were designed by Roche Diagnostics (Tokyo, Japan). The primer sequences were as follows:

MMCP-4: Forward: 5′-ggcctgtaaaaactattggcatt-3′, Reverse: 5′-cacacagtagaggtcctccaga-3′;

Collagen I: Forward: 5′-catgttcagctttgtggacct-3′, Reverse: 5′-gcagctgattgagggatgt-3′;

TGF-β1: Forward: 5′-tggagcaacatgtggaactc-3′, Reverse: 5′-cagcagccggttaccaag-3′;

18S rRNA: Forward: 5′-gcaattattccccatgaacg-3′, Reverse: 5′-gggacttaatcaacgcaagc-3′.

The probe sequences were as follows:

MMCP-4: 5′-tccaggtc-3′;

Collagen I: 5′-tcctgtc-3′;

TGF-β1: 5′-ttcctggc-3′;

18S rRNA: 5′-ttcccagt-3′.

The mRNA levels of MMCP-4, collagen I, and TGF-β1 were normalized to those of 18S rRNA.

### 4.5. Histological Studies

Carnoy-fixed kidneys were embedded in paraffin, and 3-μm-thick serial cross-sections were prepared using a sliding microtome (LITORATOMU, REM-710, Yamato Koki Kogyo Ltd., Asagiri, Saitama, Japan). To assess renal fibrosis, the first section from each sample was stained with Azan–Mallory stain. Three areas per section were randomly selected at 200× magnification using a computerized morphometry system (NIS-Elements Documentation, v.3.07, Nikon, Tokyo, Japan). The average fibrotic area was then quantified using WinROOF2021, an image analysis software (MITANI Corporation, Tokyo, Japan). To evaluate mast cell distribution, the second section from each sample was stained with toluidine blue. Briefly, after deparaffinization with remosol (Wako Pure Chemicals, Osaka, Japan), sections were immersed in a 0.5% toluidine blue solution (pH 4.8) for approximately 15 min, fractionated with a 0.5% glacial acetic acid solution, and mounted after drying. To determine chymase distribution, immunohistochemical staining was performed on the third serial section using an anti-MMCP-4 antibody (ab92368, Abcam, Cambridge, UK), following a previously described protocol [36]. Briefly, to suppress endogenous peroxidase activity and nonspecific binding, deparaffinized sections were sequentially incubated with 3% hydrogen peroxide and a protein-blocking solution, each for 5 min at room temperature. The sections were then incubated overnight at 4 °C with the diluted primary antibody (1:100), followed by detection using a labeled streptavidin-biotin peroxidase kit (Dako LSAB kit, Carpinteria, CA, USA) and 3-amino-9-ethylcarbazole for color development. Sections were lightly counterstained with hematoxylin and mounted with cover glasses. Following staining, mast cell and chymase-positive cell counts were evaluated under a computerized morphometry system, and the number of cells per unit area was compared between experimental groups.

### 4.6. Statistical Analysis

All numerical data are expressed as the mean ± standard error of the mean (SEM). Statistically significant differences among multiple groups were assessed using one-way analysis of variance (ANOVA), followed by post hoc analysis (Fisher’s test). A *p* value of <0.05 was considered statistically significant. Correlations were analyzed using Pearson’s correlation coefficient.

## 5. Conclusions

In conclusion, chymase gene expression, the number of chymase-positive mast cells, and the fibrosis-related factor TGF-β1 were all significantly elevated in kidneys with I/R injury. Treatment with the chymase-specific inhibitor TY-51469 reduced chymase expression and TGF-β1 levels, resulting in a significant improvement in fibrosis formation. These findings indicate that mast cell-derived chymase plays a role in the pathogenesis of renal fibrosis following I/R injury in mice. Furthermore, the inhibition of chymase might represent a potential therapeutic strategy to prevent the transition from AKI to CKD.

## Figures and Tables

**Figure 1 ijms-26-03913-f001:**
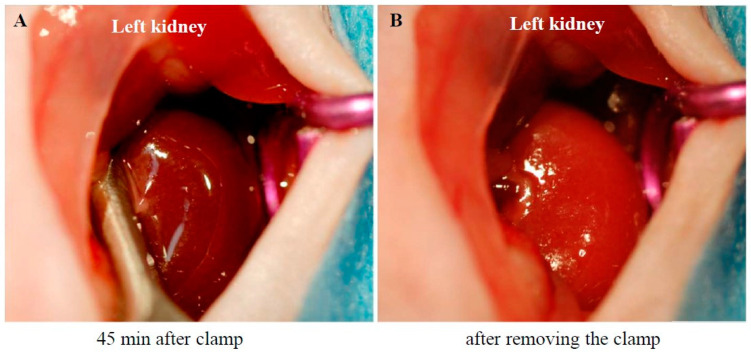
Changes in kidney surface color during ischemia (**A**) and reperfusion (**B**).

**Figure 2 ijms-26-03913-f002:**
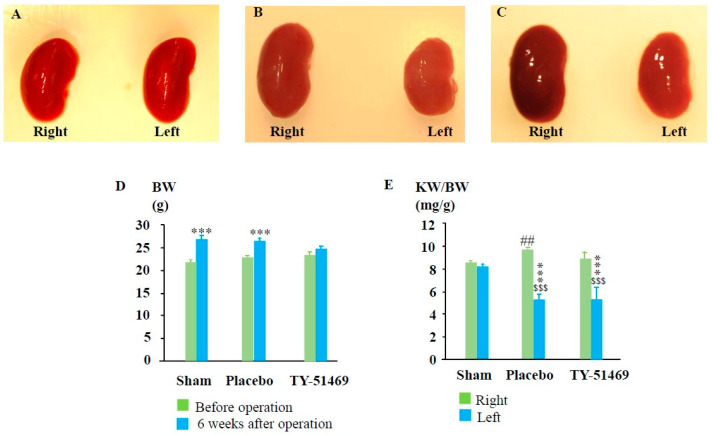
Changes in BW and KW 6 weeks after the sham operation or I/R injury. Representative photographs of the right and left kidneys 6 weeks after surgery from a sham-operated mouse (**A**), and from I/R mice treated with a placebo (**B**) and TY-51469 (**C**). (**D**) Changes in BW 6 weeks after the sham operation or after the I/R injury. (**E**) Changes in the ratio of KW to BW 6 weeks after the sham operation or I/R. ^$$$^
*p* < 0.001 vs. the non-ischemic contralateral kidney. ^##^
*p* < 0.01 vs. the sham-operated right kidney. *** *p* < 0.001 vs. the sham-operated left kidney.

**Figure 3 ijms-26-03913-f003:**
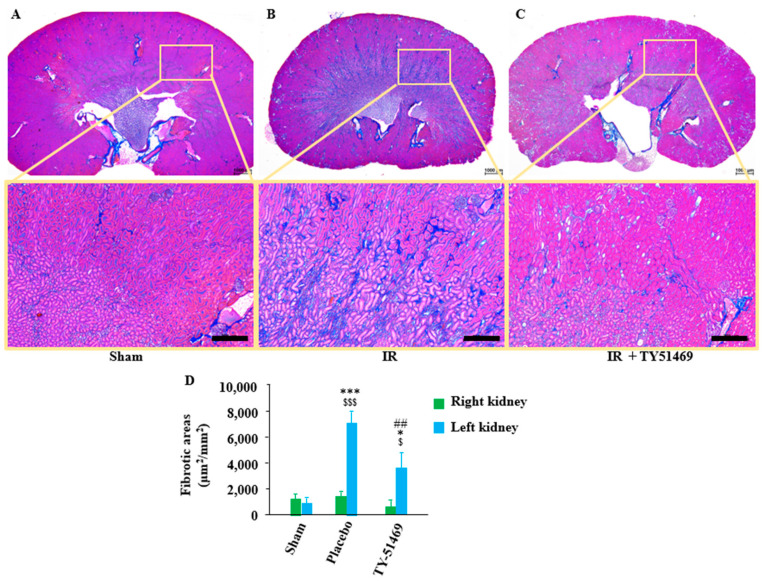
Azan–Mallory staining and quantification of renal fibrotic areas. Representative images of Azan–Mallory-stained cross-sections from a sham-operated mouse (**A**) and from an I/R mouse treated with a placebo (**B**) and TY-51469 (**C**). The blue-stained areas represent collagen deposition. (**D**) Quantification of renal interstitial fibrotic areas in cross-sections of kidneys from sham-operated mice or I/R mice treated with a placebo and TY-51469. ^$^
*p* < 0.05, ^$$$^
*p* < 0.001 vs. the non-ischemic contralateral kidney. Bar indicates 200 μm. * *p* < 0.05, *** *p* < 0.001 vs. the sham-operated left kidney. ^##^
*p* < 0.01 vs. the placebo-treated left kidney.

**Figure 4 ijms-26-03913-f004:**
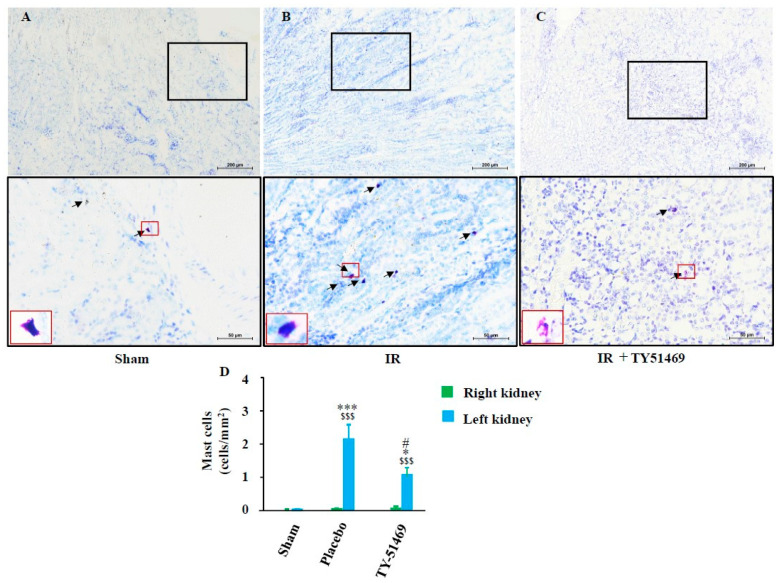
Toluidine blue staining and quantification of mast cells. Representative images of toluidine blue-stained cross-sections of the kidneys of a sham-operated mouse (**A**) and I/R mouse treated with a placebo (**B**) and TY-51469 (**C**). Mast cells are indicated by black arrows. (**D**) Quantification of mast cells in cross-sections of kidneys from sham-operated mice and I/R mice treated with a placebo and TY-51469. ^$$$^
*p* < 0.001 vs. the non-ischemic contralateral kidney. * *p* < 0.05, *** *p* < 0.001 vs. the sham-operated left kidney. ^#^
*p* < 0.05 vs. the placebo-treated left kidney.

**Figure 5 ijms-26-03913-f005:**
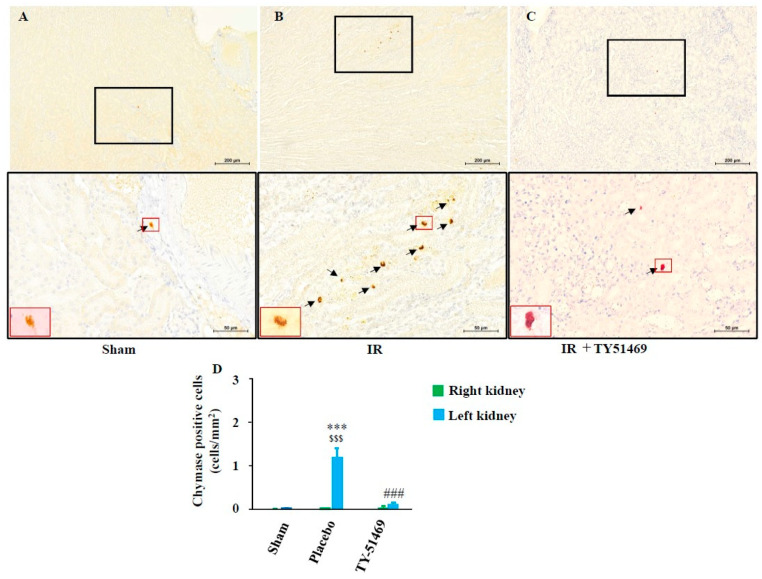
Chymase immunostaining and quantification of chymase-positive cells. Representative images of chymase-immunostained cross-sections from a sham-operated mouse (**A**) and from I/R mice treated with a placebo (**B**) and TY-51469 (**C**). Chymase-positive cells are indicated by black arrows. (**D**) Quantification of chymase-positive cells in cross-sections of kidneys from sham-operated mice and I/R mice treated with a placebo and TY-51469. ^$$$^
*p* < 0.001 vs. the non-ischemic contralateral kidney. *** *p* < 0.001 vs. the sham-operated left kidney. ^###^
*p* < 0.001 vs. the placebo-treated left kidney.

**Figure 6 ijms-26-03913-f006:**
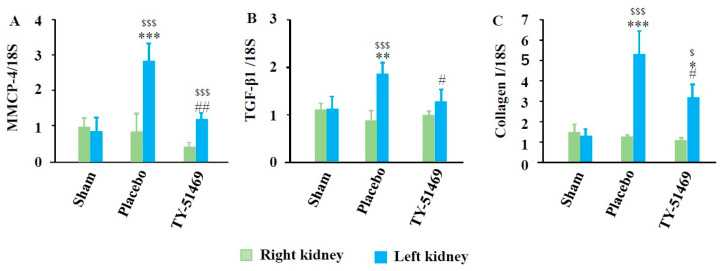
Changes in MMCP-4 (**A**), TGF-β1 (**B**), and collagen I (**C**), mRNA expression 6 weeks after the sham operation or I/R. ^$^
*p* < 0.05, ^$$$^
*p* < 0.001 vs. the non-ischemic contralateral kidney. * *p* < 0.05, ** *p* < 0.01, *** *p* < 0.001 vs. the sham-operated left kidney. ^#^
*p* < 0.05, ^##^
*p* < 0.01 vs. the placebo-treated left kidney.

**Figure 7 ijms-26-03913-f007:**
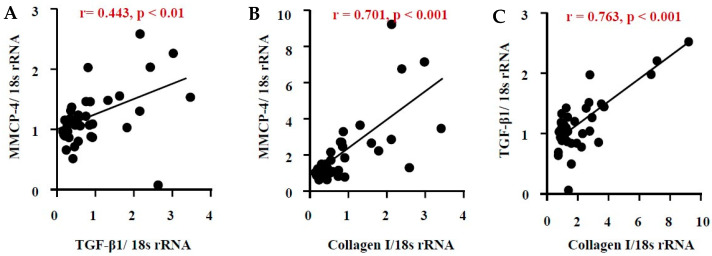
Linear regression analyses of mRNA expression levels of MMCP-4, TGF-β1, and collagen I. Correlations were observed between MMCP-4 and TGF-β1 mRNA levels (**A**), MMCP-4 and collagen I mRNA levels (**B**), and TGF-β1 and collagen I mRNA levels (**C**) in all the examined kidneys 6 weeks after surgery.

## Data Availability

The data presented in this study are available on request from the corresponding author.

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
