# Peer review of "Chymase Inhibition Attenuates Kidney Fibrosis in a Chronic Mouse Model of Renal Ischemia–Reperfusion Injury"

_ijms, 2025, doi:10.3390/ijms26083913_

Round 1

Reviewer 1 Report

Comments and Suggestions for Authors

This manuscript presents a well-structured and thorough investigation into the role of chymase inhibition in preventing renal fibrosis following ischemia-reperfusion (I/R) injury in a mouse model. The study is well-conceived, employing appropriate experimental techniques to assess the impact of the chymase inhibitor TY-51469. The findings provide significant insights into potential therapeutic strategies for preventing the transition from acute kidney injury (AKI) to chronic kidney disease (CKD).

Areas for Improvement:

  1. The paragraph stating, “Endothelial dysfunction can lead to oxygen deficiency in kidney tissue, resulting in mitochondrial dysfunction in various cell types and damage to renal tubular epithelial cells, ultimately triggering sterile interstitial inflammation. Simultaneously, repair mechanisms are activated in response to this inflammation. However, excessive repair following AKI can lead to renal fibrosis, further damaging normal kidney structure and promoting the progression from AKI to CKD.” (lines 51–57) must include references.
  2. The section "2.2 Characteristics of renal fibrosis after I/R" lacks clarity and should be rewritten to enhance readability and logical flow.
  3. To strengthen the conclusions, a western blot analysis for MMCP-4, TGF-β1, and collagen I should be included. This addition would validate whether transcriptional changes observed in the study correspond to actual protein expression levels, as transcriptional effects do not always translate into protein-level changes.
  4. The manuscript should discuss the application of chymase inhibitors (not necessary TY-51469) in other diseases and their safety profile, including potential side effects.
  5. The authors should elaborate on the advantages of chymase inhibitors compared to other available treatments for AKI. 
  6. The limitations of the study should be clearly discussed.

Author Response

April 15, 2025

Dear Dr. reviewer 1

Thank you very much for your valuable comments. In accordance with your comments, we have modified our manuscript as follows;

  1. Based on your instructions, we have cited two new references to support the statements made in lines 58–64 of the text.

  1. As you rightly pointed out, the previous description related to kidney fibrosis might have been difficult for readers to understand. Therefore, we have added the following description to the Results section (Lines 139~145):

As shown in Figure 3A, Azan-Mallory staining of the sham-operated kidney revealed normal histology with no significant renal fibrosis. As is well known, the Azan-Mallory staining method stains collagen fibers within fibrous connective tissue blue. Since collagen fibers are commonly distributed around blood vessels, these areas typically appear blue with Azan-Mallory staining, even in normal kidneys. In Figure 3A, no obvious blue staining was observed in the sham-operated kidney, aside from the tissue surrounding blood vessels.

  1. We are very grateful for your constructive feedback. It is indeed true that increased gene expression of MMCP-4, TGF-β1, and collagen I does not necessarily reflect increased protein levels. Ideally, we should have investigated whether the elevated mRNA expression of these genes correlates with their corresponding protein levels. However, due to the limited time available for this revision, we regret that we are unable to address this point in the present study. We have acknowledged this as a limitation of our research in the revised manuscript (Lines 315–319).

Limitation: In the present study, increased expression of MMCP-4, TGF-β1, and collagen I was observed at the mRNA level. However, as mRNA expression does not necessarily correlate with protein expression, the relationship between the mRNA and protein levels of these molecules remains unclear. Further studies are needed to clarify whether the observed changes in gene expression are reflected at the protein level.

  1. Thank you for your helpful comments. To date, we have investigated the effects of various chymase inhibitors in animal models of different diseases. We previously examined the effects of orally administered chymase inhibitors, such as NK3201 (Jin D, Takai S, Yamada M, Sakaguchi M, Kamoshita K, Ishida K, Sukenaga Y, Miyazaki M. Impact of chymase inhibitor on cardiac function and survival after myocardial infarction. Cardiovasc Res. 2003;60:413–420) and BCEAB (Jin D, Takai S, Yamada M, Sakaguchi M, Miyazaki M. Beneficial effects of cardiac chymase inhibition during the acute phase of myocardial infarction. Life Sci. 2002;71:437–446) in the pathology of myocardial infarction in hamster models. In the present study, we employed the chymase inhibitor, TY-51469. However, since we did not provide much information about TY-51469 in this article, we have added the following information about this compound in the Introduction section of the revised paper (Lines 84~90). Additionally, since TY-51469 has not been used in clinical trials, its side effects are currently unknown. However, no obvious side effects have been found in the data from animal experiments so far.

The acute management of AKI primarily involves symptomatic treatment of complications, such as pulmonary edema and hyperkalemia. Chymase inhibitors are unlikely to offer immediate benefits in this setting. Currently, no therapeutic agents have been definitively established to prevent the progression from AKI to CKD. While ACE inhibitors are clinically used in the treatment of diabetic nephropathy, they might also help suppress the transition from AKI to CKD by inhibiting Ang II production. In comparison, chymase inhibitors not only suppress Ang II generation, but also inhibit the activation of profibrotic factors, such as TGF-β1. This dual mechanism suggests that chymase inhibitors might offer superior therapeutic efficacy compared to ACE inhibitors in attenuating the progression from AKI to CKD.

Another limitation of this study is the exclusive focus on the 6-week endpoint. We recognize that earlier assessment of chymase activity and inflammatory cytokine levels following I/R injury would have provided a more comprehensive understanding of chymase's pathophysiological role. We plan to address these aspects in future studies.

Thank you again for your helpful comments. We fell that our present paper is become more mature with this revision!

Thank you very much for your kind consideration!

With best regards,

Denan Jin, M.D., Ph.D.

Department of pharmacolog, Osaka Medical and Pharmaceutical University, 2-7 Daigaku-machi, Takatsuki, Osaka 569-8686, Japan.

                     TEL: +81-72-683-1221 (Ext2141)

                     FAX: +81-72-684-6730

                     E-mail: denan.jin@ompu.ac.jp

Reviewer 2 Report

Comments and Suggestions for Authors

The study is novel. The experiments were designed and conduced well. I have following suggestion for improvement:

  1. Please give the full name of abbreviation (such as KW and BW) at the first time. Please introduce  TY-51469 more and the rationale to use the inhibitor.
  2.  Please provide concise rationale and description  of experiment design before directly conclude the results.
  3. In addition to MMCP-4, did you also examine  MMCP-1, MMCP-2, MMCP-3? Why were the number of mast cells and MMCP-4-240 positive cells in the placebo-treated ischemic  kidney? Please at least prove some explanation in discussion.
Comments on the Quality of English Language

The English is Ok, but concise rationale and description  of experiment design before results is strongly suggested.

Author Response

April 15, 2025

Dear Dr. reviewer 2

Thank you very much for your valuable comments. In accordance with your following comments, we have modified our manuscript as follows;

 (Lines305~314). The acute management of AKI primarily involves symptomatic treatment of complications, such as pulmonary edema and hyperkalemia. Chymase inhibitors are unlikely to offer immediate benefits in this setting. Currently, no therapeutic agents have been definitively established to prevent the progression from AKI to CKD. While ACE inhibitors are clinically used in the treatment of diabetic nephropathy, they might also help suppress the transition from AKI to CKD by inhibiting Ang II production. In comparison, chymase inhibitors not only suppress Ang II generation, but also inhibit the activation of profibrotic factors, such as TGF-β1. This dual mechanism suggests that chymase inhibitors might offer superior therapeutic efficacy compared to ACE inhibitors in attenuating the progression from AKI to CKD.

As shown in Figure 3A, Azan-Mallory staining of the sham-operated kidney revealed normal histology with no significant renal fibrosis. As is well known, the Azan-Mallory staining method stains collagen fibers within fibrous connective tissue blue. Since collagen fibers are commonly distributed around blood vessels, these areas typically appear blue with Azan-Mallory staining, even in normal kidneys. In Figure 3A, no obvious blue staining was observed in the sham-operated kidney, aside from the tissue surrounding blood vessels.

  • The following statement has also been added regarding the assessment of mast cells by toluidine blue staining (Lines 167–170):

Toluidine blue staining is a simple and traditional method used to identify mast cells. The cytoplasm of mast cells exhibits metachromatic staining, appearing red-purple, which makes them easily distinguishable from other cell types.

Thank you again for your helpful comments. We fell that our present paper is become more mature with this revision!

Thank you very much for your kind consideration!

With best regards,

Denan Jin, M.D., Ph.D.

Department of pharmacolog, Osaka Medical and Pharmaceutical University, 2-7 Daigaku-machi, Takatsuki, Osaka 569-8686, Japan.

                     TEL: +81-72-683-1221 (Ext2141)

                     FAX: +81-72-684-6730

                     E-mail: denan.jin@ompu.ac.jp

Round 2

Reviewer 1 Report

Comments and Suggestions for Authors

The authors have responded to my concerns.
While it would be preferable to include western blot analyses for MMCP-4, TGF-β1, and collagen I, the Azan-Mallory staining shown in Figure 3 detects collagen, supporting the notion that the increased transcription of collagen I may correspond to elevated protein expression. Similarly, for MMCP-4, the immunohistochemistry results in Figure 5 show increased expression in the IR kidney.